# Intussusceptive Angiogenesis and Peg–Socket Junctions between Endothelial Cells and Smooth Muscle Cells in Early Arterial Intimal Thickening

**DOI:** 10.3390/ijms21218049

**Published:** 2020-10-28

**Authors:** Lucio Díaz-Flores, Ricardo Gutiérrez, Mª Pino García, Sara Gayoso, José Luís Carrasco, Lucio. Díaz-Flores, Miriam González-Gómez, Juan Francisco Madrid

**Affiliations:** 1Department of Basic Medical Sciences, Faculty of Medicine, University of La Laguna, 38071 Tenerife, Spain; histologia54@gmail.com (R.G.); pilargon59@gmail.com (S.G.); jcarraju@ull.edu.es (J.L.C.); ldfvmri@yahoo.com (L.D.-F.J.); mirgon@ull.edu.es (M.G.-G.); 2Department of Pathology, Eurofins® Megalab–Hospiten Hospitals, 38100 Tenerife, Spain; mpgarcias@megalab.es; 3Department of Cell Biology and Histology, School of Medicine, Campus of International Excellence “Campus Mare Nostrum”, IMIB-Arrixaca, University of Murcia, 30100 Murcia, Spain; jfmadrid@um.es

**Keywords:** arterial intimal thickening, angiogenesis, intussusceptive angiogenesis, peg-and-socket junctions, vascular regression, endothelial cells, vascular smooth muscle cells, platelets

## Abstract

Angiogenesis in arterial intimal thickening (AIT) has been considered mainly in late AIT stages and only refers to sprouting angiogenesis. We assess angiogenesis during early AIT development and the occurrence of the intussusceptive type. For this purpose, we studied AIT development in (a) human arteries with vasculitis in gallbladders with acute cholecystitis and urgent (*n* = 25) or delayed (*n* = 20) cholecystectomy, using immunohistochemical techniques and (b) experimentally occluded arterial segments (*n* = 56), using semithin and ultrathin sections and electron microscopy. The results showed transitory angiogenic phenomena, with formation of an important microvasculature, followed by vessel regression. In addition to the sequential description of angiogenic and regressive findings, we mainly contribute (a) formation of intravascular pillars (hallmarks of intussusception) during angiogenesis and vessel regression and (b) morphological interrelation between endothelial cells (ECs) in the arterial wall and vascular smooth muscle cells (VSMCs), which adopt a pericytic arrangement and establish peg-and-socket junctions with ECs. In conclusion, angiogenesis and vessel regression play an important role in AIT development in the conditions studied, with participation of intussusceptive angiogenesis during the formation and regression of a provisional microvasculature and with morphologic interrelation between ECs and VSMCs.

## 1. Introduction

Arterial intimal thickening (AIT) is a process observed in normal and pathological conditions, characterized by the presence of neointimal (myointimal) cells and newly formed extracellular matrix between the endothelium and the internal elastic lamina of the artery. In pathology, AIT participates in the development of occlusive vascular diseases, such as atherosclerosis, bypass stenosis, and restenosis. Atherosclerosis, the pathologic process with major morbidity and mortality, has its morphological substrate in the atheromatous plaque, which leads to artery hardening and narrowing. Plaque formation in atherosclerosis involves intertwined pathologic pathways and endothelial dysfunction, prothrombotic state, and vascular inflammation [1,2].

Angiogenesis in AIT has been demonstrated mainly in lesion progression and destabilization [3,4,5,6,7]. However, angiogenic events and types of angiogenesis in early stages of AIT development have been poorly studied: In humans, because AIT is generally found in advanced or complicated stages when the lesion is surgically removed; and in experimental conditions, because neovascularization has been mainly explored by considering adventitial capillary density [8]. Indeed, arterial neovascularization is rapidly (in days) followed by regression of most newly formed vessels [9], except those in the adventitia. Therefore, an immunohistochemical and ultrastructural study of the arterial microvasculature and its evolution in early stages of AIT formation is appropriate to identify angiogenic and vessel regressive findings during lesion development. 

Sprouting angiogenesis and intussusceptive angiogenesis are the two principal and complementary forms of angiogenesis, with synergistic interaction [10,11,12,13,14,15,16]. Sprouting angiogenesis is defined as a multistep complex process of neovascularization by the sprouting of capillaries from pre-existing vessels. This process includes EC migration, extracellular matrix changes, EC proliferation, pericyte mobilization, tubulogenesis (vascular lumen development), formation of a new basal membrane, and sprouting connection. Inflammatory cells also participate in this process. Intussusceptive angiogenesis is the mechanism by which pre-existing vessels split, expand and remodel through transluminal pillar formation, contributing to microvascular growth, vessel arborization, branching remodeling and vessel segmentation [10]. 

In AIT, sprouting angiogenesis has been described, whereas the occurrence of intussusceptive angiogenesis has not been taken into consideration. The study of intussusceptive angiogenesis is of great interest during AIT formation because it participates in microvascular growth, and in morphogenesis and remodeling [17,18,19,20,21,22,23,24,25,26,27,28,29]. For intimal neovascularization to occur, the adventitial microvasculature must cross the media layer of the artery. For this reason, the possible interrelationship between endothelial cells (ECs) and vascular smooth muscle cells (VSMCs) should also be explored.

Gallbladder vasculitis can be part of systemic vasculitis or focal single-organ vasculitis [30,31]. In addition to arteritis, early AIT may occur in occluded muscular arteries of gallbladder in acute cholecystitis, in which one treatment is urgent cholecystectomy. A study to assess the initial and evolutive morphologic findings of microvascularization in arteries of gallbladders surgically removed by acute cholecystitis is therefore appropriate to understand angiogenic and vessel regressive phenomena during AIT formation in humans. However, the detection of arteries with AIT in early stages of development should be carried out in paraffin embedded tissue and by observing extensive regions of the gallbladder, for which conventional and immunohistochemical procedures are the most indicated. Likewise, intravascular pillar (hallmark of intussusceptive angiogenesis) identification requires 3D demonstration or serial semithin or ultrathin sections [12,13,18,19,23,24,32,33,34,35] and the interrelationship between ECs and VSMCs is most precise to demonstrate in ultrastructural observation. Therefore, a complementary study under electron microscopy and in serial semithin sections in experimentally occluded arterial segments would improve the study.

Given the above, the objectives of this study are to assess (a) the morphological angiogenic events that occur in the initial stages of AIT development in medium- and small-caliber arteries, (b) the possible participation of intussusceptive angiogenesis during AIT development, and (c) the morphologic interrelationship between ECs in the arterial media layer and VSMCs. To these ends, we will explore (a) the human arteries of gallbladders surgically removed for acute cholecystitis, using conventional and immunohistochemical procedures and (b) occluded segments of rat femoral artery, using electron microscopy and serial semithin sections.

## 2. Results

### 2.1. Angiogenesis in Arteries of the Human Gallbladder with Vasculitis and AIT Formation in Acute Cholecystitis

#### 2.1.1. Arteries in Gallbladders with Acute Cholecystitis and Urgent Cholecystectomy

In cases with early removal of the gallbladder with acute cholecystitis and arterial involvement, some arteries showed the following events: Luminal and parietal inflammatory infiltrate (neutrophils, lymphocytes, and macrophages) and loss of the endothelium, parietal and periarterial hemorrhage, and intraluminal and/or parietal fibrin deposits (Figure 1A,B). In addition, intimal thickening development was frequently observed, including angiogenic findings, leading to arterial microvascularization. Next, we will pay particular attention to the microvasculature in these arteries in which intimal thickening develops.

Numerous microvessels were observed in some zones of the arterial adventitial layer (Figure 1C,D). The capillaries in the adventitia showed endothelial cells (ECs) and pericytes (Figure 1C,D). When the microvessel ECs are in the external part of the muscular layer of the artery, the capillaries appear to be lined by VSMCs, which adopt a pericytic aspect (Figure 1E). Interendothelial contacts and EC bridges were observed with relative frequency in the microvessels (Figure 1F–I).

Capillaries in the media layer were also observed, above all in arteries with fewer vascular muscle cells (Figure 2A). When the capillaries are present in the arterial lumen (Figure 2B), their ECs are observed between thrombus components and the VSMCs appear underlying the internal elastic membrane. Several lumens lined by ECs and with a varying number of red blood cells were also present underlying these VSMCs (Figure 2C). EC bridges were present in these intra-arterial neolumens (Figure 2D).

The neovessels underlying the myointimal cells presented ECs and αSMA+ cells with a pericytic aspect, and frequently originated complex networks (Figure 2E), in which numerous vessel loops were formed (Figure 2E,F). These vessel loops surrounded interstitial tissue structures (ITSs) including the αSMA cells present on the innermost side of the vessel loop (Figure 2E,F). Therefore, in these arteries, the region in which intimal thickening begins varies in amplitude and lies between the media layer of the artery and the newly formed vessel networks (Figure 3A).

ITSs covered by ECs, suggesting intraluminal large pillars, were present in vessel loops in which their lumen appeared partially or totally open (Figure 3B–H). Thus, the pillar-like structure showed a cover and a core. The cover was formed by CD34+ ECs, which corresponded to the vessel loop inner layer, whereas the core was the ITS. Small pillar-like structures formed by extensions of ECs were also seen (Figure 3E,F). The pillar-like structures could be partially joined, adopting variegated forms (Figure 3H).

#### 2.1.2. Arteries in Acute Cholecystitis with Delayed Cholecystectomy

Arteries with an advanced stage of intimal thickening formation were identified, although it was still possible to observe pillar-like structures in a newly formed central lumen (Figure 4A). In some arteries, numerous capillaries were present in zones of transition between the adventitia and the media layer (Figure 4A,B) and underlying the internal elastic lamina (Figure 4B). Occasionally, persistent vessels were observed all the way from the adventitia to areas underlying the internal elastic lamina (Figure 4C). Numerous CD34+ filopodia-like in ECs were seen in some of these microvessels (Figure 4D–F).

Numerous intravascular pillar-like structures were observed in persistent vessels in the intimal thickening (Figure 4G). Most of these structures were formed by ECs bridging the vascular lumen (Figure 4G). Well-developed AIT with myointimal cells arranged in several layers was finally seen in some arteries of the gallbladders with acute cholecystitis and delayed cholecystectomy (Figure 4H–J)

Arteries in unaffected human gallbladders were within normal limits.

### 2.2. Angiogenesis in Experimental Occluded Arterial Segments with Intimal Thickening Formation

On days 1–2, loss of the endothelium was observed in the occluded arterial segments (Figure 5A). By day three/four, capillaries were present through the artery wall (Figure 5B). The ECs in the media layer appeared surrounded by VSMCs (Figure 5C) and globular structures covered by a plasmatic membrane and with abundant filaments (Figure 5D). Although the globular structures appeared frequently isolated, in other ultrathin sections they were shown to correspond to vascular smooth muscle cytoplasmic processes, which widened in their terminal portions (bulb-shaped protrusions) (Figure 5E,F). These protrusions established a close association between VSMCs and ECs by heterocellular peg-and-socket-type junctions. Thus, the VSMCs originated the pegs (bulb-shaped protrusion), while the ECs formed the sockets in their abluminal surfaces (Figure 5E). Vascular smooth muscle protrusions reaching the capillary lumen were also observed (Figure 5F). The perivascular SMCs were prominent (Figure 6A) and the ECs showed increased cell volume, a vesicular nucleus and a large cytoplasm, which contained numerous ribosomes, either isolated or aggregated. Intracytoplasmic stress fibers were also seen in the ECs (Figure 6B and insert).

On days four to six, capillaries were observed through the internal elastic membrane, which showed some discontinuities and folds (Figure 6C). The arterial lumen was occupied by abundant red blood cells, fibrin, and myriad platelets and leukocytes (predominantly macrophages and eosinophils) (Figure 6D,E). Intraluminal accumulations of ECs were observed in association with the capillaries and in proximity to discontinuities and folds of the internal elastic membrane (Figure 6C). Frequently, EC aggregates, located between the red blood cell, fibrin and platelet components, acquired a spheroid aspect (Figure 6D,E). In these spheroids, the ECs showed an extensive cytoplasm, a vesicular nucleus, prominent nucleoli, and some figures of mitosis (Figure 6D–F). In the areas with discontinuities of the internal elastic membrane, where capillaries appear through the internal elastic membrane, EC spheroids were present in the artery lumen. VSMCs also were seen on the luminal surface of the internal elastic membrane (Figure 6C) and occasionally on the endothelial spheroids.

The multicellular spheroids reorganized with luminal and abluminal EC polarization. ECs emerging from the spheroids were observed over the luminal surface of the internal elastic membrane, over the myointimal cells and into the arterial lumen, originating nascent pillars (Figure 7A). ECs in these nascent pillars were seen in association with platelets, forming peg-and-socket junctions (platelets formed the peg and ECs formed the socket) (Figure 7B and insert).

EC bilayers originated intraluminal early pillars, in which the abluminal surface of ECs faced each other. Fibrin deposits, platelets, scarce perivascular cells and occasional red blood cells and macrophages were present in spaces between the abluminal surface of EC bilayers. These incarcerated components formed an initial core or ITS (Figure 7C,D). Thus, the ITSs (the core) and the surrounding ECs (the cover) formed the intussusceptive transluminal pillars bridging opposite walls in the lumen of the occluded arterial segment. From these pillars, new nascent pillars were also seen (Figure 7D). Appearance and disappearance, and continuities and discontinuities of the nascent and early pillars were observed in serial semithin sections (Figure 8A–D).

Subsequently (days 6–8), myointimal cells were present in the ITSs of mature pillars (Figure 8E,F). Thus, the mature pillars presented a cover formed by ECs and a core formed by myointimal cells and extracellular matrix. From these pillars, ECs forming nascent pillars were still seen (Figure 8E). Appearance and disappearance, and continuities and discontinuities of these mature pillars were also observed (Figure 9A–C). The resulting capillary network with numerous vessel loops fused with other capillary networks originated from adjacent spheroids.

On days 8 to 12, vessel spaces with wider lumens converged in the axial region of the artery, and a newly formed preferential vessel with a path parallel to the longitudinal axis of the artery was present (Figure 9D). Simultaneously, most of the microvessels that formed the intra-arterial networks regressed. The myointimal cells that originally grew on the luminal surface of the internal elastic membrane and those in the regressive vessels and in their ITS pillars formed several layers (increase in intimal thickening). Therefore, intussusception with regressive phenomena was present in the vessel loops. These regressive phenomena included (a) formation of numerous endothelial contacts, (b) perforation of the contacts, with vessel loop fragmentation in capillary-sized vessels, which showed numerous red blood cells, platelets and cellular debris in their lumens (Figure 9D,E) and (c) loss of ECs and persistence of myointimal cells, which also contributed to intimal thickening. The appearance and disappearance of endothelial contacts and their perforations, as well as the fragmentation of the loops in regressive capillary-like were also observed in serial semithin sections (Figure 10). However, some vessel loops persisted between the preferential vessel and the internal elastic membrane. Finally, on days 13 and 14, similar images to those of well-formed AIT in arteries of human gallbladder with acute cholecystitis were observed in experimental conditions. 

Normal and sham-operated femoral arteries were unmodified, except for the presence of a few neutrophils and mononuclear cells in the surrounding tissue of the sham-operated cases.

## 3. Discussion

The facts in this work show that angiogenesis is one of the fundamental mechanisms in the early development of intimal thickening in the conditions studied. The observation of early intimal thickening development in human pathology and experimentally has allowed us to follow the sequence of events and to demonstrate (a) the participation of intussusceptive angiogenesis in intimal thickening development and in the regression of most of the newly formed microvasculature and (b) peg and socket junctions between ECs and VSMCs in the media layer of the artery.

Successive and overlapping main events in intimal thickening formation in arteries with vasculitis or experimentally occluded can be summarized as follows: (1) Intimal EC loos in both conditions, to which is added vasculitis with hemorrhage and fibrin deposits in the gallbladder with acute cholecystitis. (2) Increased adventitial microvascularization. (3) ECs establishing morphologic relationship with VSNCs in the media layer of the artery. (4) VSMCs and vessels through the arterial internal elastic lamina give rise to (a) one or several layers of myointimal cells on the luminal surface of the internal elastic lamina (early intimal thickening), (b) deposits of fibrinous material, myriad platelets, red blood cells and macrophages in the lumen of the artery and (c) aggregates of ECs, forming spheroids within these deposits. (5) Early intraluminal intussusceptive pillars. (6) Vascular networks with mature pillars. (7) A centric or eccentric preferential vessel and vascular network regression with late pillars. (8) mature intimal thickening with occasional regressive vessels. Of these facts, we will briefly discuss those already known and more widely our original contributions in this study.

One of the main triggers of AIT is intimal EC denudation, which can be due to several causes, including ischemia and inflammation [8,36,37,38,39]. In our study, ischemia in the experimentally occluded arterial segments and vasculitis in the gallbladder arteries are the likely main causes of endothelium loss. The increased adventitial microvasculature has been observed by several authors in intimal thickening and related with VEGF [40].

One of our original contributions is the morphologic interrelation between ECs and VSMCs in the media layer of the artery. Indeed, VSMCs acquire a pericytic disposition around ECs and contribute to form peg-and-socket junctions between both cell types (VSMCs form the peg and ECs form the socket). Peg-and-socket junctions are established between pericytes and ECs [41,42,43,44]. Our group has also described their presence between VSMCs and ECs in myopericytomas [45] and experimentally in veins after perivenous administration of prostaglandins and glycerol [46].

Our demonstration of EC spheroids (above all in occluded arterial segments) in a fibrin–platelet substrate and other blood components highlight an in vivo complex process in a natural environment, which has its counterpart in the experimental models that use EC spheroids in angiogenesis research and regenerative medicine [47,48,49]. This similarity is increased when using three-dimensional cocultures of ECs–VSMCs separated by flexible, porous polydimethylsiloxane membranes mimicking the porosity of the internal elastic membrane [50].

The demonstration of intravascular pillars in the lumen of the artery and of the newly formed vessels allows us to answer one of the main questions raised in this work: Whether intussusceptive angiogenesis participates in the development and evolution of AIT. Indeed, pillars are hallmarks of intussusceptive angiogenesis [8,10,12,17,18,19,24,25,27,28,29,32,33,51,52,53,54,55,56]. In our study, pillars show the cover and core of these structures and meet the requirement of subsequent appearance and disappearance in serial semithin sections [33]. Intussusceptive angiogenesis can therefore be complementary to the sprouting angiogenesis previously described in intimal thickening.

Several structures and mechanisms are involved in pillar formation, including endothelial ridges and transluminal interendothelial bridges (nascent pillars), endothelial contacts of opposite vessel walls, merged adjacent capillaries with modified contacting cells, vessel loops, and split pillars [19,21,28,29,32,34,35,57]. In our study, the observation of pillars in successive times, with variable associated structures and ITS components suggests the predominance of a different mechanism, depending on the evolutive stage of AIT.

Pillars immediately formed after penetration of ECs in the arterial lumen appear to arise from endothelial bridges, originating intraluminal spaces in the artery. The demonstration of frequent peg-and-socket junctions between platelets and bridging ECs (the platelets form the peg and the ECs form the socket) is also a fact not previously reported. The early penetration of microvessels through the arterial wall providing fibrinous material and myriad platelets in the lumen of the artery creates a medium where the following phenomena develop. In an analysis of the cross-talk between ECs and VSMCs a pronounced activation was observed when activated platelets were added [58]. The myriad platelets in the lumen of the occluded artery and in newly formed microvessels, as well as in the interstitial structures, can increase these interactions and the myointimal response.

Pillars in the newly formed intra-arterial microvasculature arise from vessel loops, which surround ITSs, including myointimal cells, which acquire a pericytic location. Finally, pillars that occur in regressing vessel loops appear to form from the perforation of endothelial contacts of the opposite vessel walls of the loops with capillary-like fragmentation [11]. The persistence of neovascularization (absence of vessel regression) during AIT development could explain some pathological processes with high arterial newly formed microvasculature, such as Kawasaki disease [59,60] and Kawasaki-like syndrome in Covid-19 disease.

Blood flow has a critical action on pillar formation and vascular adaptation [12,61,62,63,64,65]. Hemodynamic forces also influence the signaling between ECs and VSMCs [50]. The formation of a preferential vessel that connects again with the arterial circulation or with persistent vessels that cross the arterial wall also modifies the blood flow. Therefore, in addition to pillar formation, hemodynamic forces (blood flow) may also act on the alignment of the myointimal cells (parallel to the longitudinal edge of the artery) and the morphology of ECs (cobblestone morphology) in the newly formed intra-arterial preferential vessel. Likewise, several factors may participate in the behavior of ECs and VSMCs in the arterial wall [66,67,68,69,70]. Further studies are required in these fields.

The characteristics and short duration of the angiogenic and vessel regressive findings during AIT development resemble those in repair to granulation tissue. The main difference is that, in AIT, the ECs in the arterial wall interrelate with VSMCs, which adopt a pericytic arrangement and establish peg-and-socket junctions with ECs.

The main limitations of this work are complementary studies in peg-and-socket junctions and accurate time points of the pathologic processes that affect human gallbladders. The first limitation is that peg-and-socket junctions were demonstrated ultrastructurally and that the electron microscopy procedures used did not permit junction protein staining or that of other markers to identify their possible regulatory role. The other limitation was partially obviated by the selection of cases with urgent and delayed cholecystectomy, which allowed us to observe intimal thickening in the human gallbladder in initial and more advanced stages. In addition, there is a correlation of the findings in these stages with those obtained in experimental conditions in which the time points were well established.

The demonstrated participation of intussusceptive angiogenesis during AIT development may have clinical and therapeutic implications. Since sprouting and intussusception are complementary forms of angiogenesis, with synergistic interaction [8,9,10,11,12,13,14], a compensatory mechanism after inhibitory treatment of sprouting angiogenesis may occur due to intussusceptive angiogenesis (failure of treatment by an escape mechanism) [13]. Likewise, the peg-and-socket junctions between ECs and smooth muscle cells and between ECs and platelets may be the morphologic substrate of mechanical and biochemical cellular interactions. Further studies are also required in these fields.

In conclusion, we study transient angiogenesis and regression of the newly-formed vessels during AIT development in human pathology and experimentally in rats. Angiogenic and newly-formed vessel regressive findings were: Increased adventitial microvascularization, presence of ECs in the arterial wall, VSMC/EC morphologic interrelation, intraluminal aggregates of ECs forming spheroids, platelet/EC morphologic interrelation, intravascular and interstitial pillar formation, and vessel regression with mature AIT development. Our main contributions in the present work include the demonstration of intussusceptive angiogenesis (pillar formation) in several stages of AIT development, and peg-and-socket junctions between VSMCs-ECs and Platelets-ECs. 

## 4. Materials and Methods

### 4.1. Human Samples

The archives of the Department of Basic Medical Sciences (Histology and Anatomical Pathology) of the University Hospital with mature (La Laguna, La Laguna University) and Eurofilms Megalab-Hospiten Hospitals of the Canary Islands were searched for cases of acute cholecystitis for the period 2005–2019. Forty-five cases showing arteritis and intimal thickening phenomena were selected. All patients of the selected cases were Caucasian, 21 males and 24 females, ages ranging from 31 to 72, predominantly in the 5th and 6th decades of life. The samples selected were divided into two groups (a) those from patients who underwent urgent cholecystectomy (generally days of the onset of symptoms) (*n* = 25) and (b) from patients with antibiotic treatment and delay (generally weeks) of cholecystectomy (*n* = 20). Non-affected gallbladder from a previous work were used as controls [71]. Ethical approval for this study was obtained from the Ethics Committee of La Laguna University (La Laguna, Comité de Ética de la Investigación y de Bienestar Animal, 03/09/2020, CEIBA 2020-0417), including the dissociation of the samples from any information that could identify the patient. The authors therefore had no access to identifiable patient information.

### 4.2. Experimental Procedures

Adult Sprague–Dawley rats with an average weight of 300 g were used in accordance with the guidelines of the Ethics Committee of La Laguna University (La Laguna, Comité de Ética de la Investigación y de Bienestar Animal, 03/09/2020, CEIBA 2020-0417). The rats were fed standard rat chow and water unrestrictedly and maintained in pathogen-free conditions.

During surgical procedures and tissue removal, the rats were anaesthetized with Ketamine (150 mg/Kg. i.p.). Using a surgical microscope, 1.5 cm segments of femoral arteries were exposed. Ligatures with a 10/0 thread were applied to both the proximal and distal ends of the dissected femoral arterial segments, without damaging the surrounding microcirculation. A collateral present in these occluded femoral segments was also ligated at a distance of 0.3 cm of the ostium. Specimens from femoral vessels (*n* = 56) were removed daily (*n* = 4) from day 1 to 14, inclusive. Control of normal and sham-operated femoral arteries were used from a previous work [72].

### 4.3. Techniques in Conventional Light Microscopy, Immunohistochemistry, and Electron Microscopy

Specimens for conventional light microscopy were fixed in a buffered neutral 4% formaldehyde solution, embedded in paraffin and cut into 3 μm-thick sections. Sections were stained with Hematoxylin and Eosin (H&E) and Trichrome staining (Roche, Basel, Switzerland. Ref. 6521908001).

For immunochemistry, histologic sections (3 μm-thick) were attached to silanized slides. After pre-treatment for enhancement of labelling, the sections were blocked with 3% hydrogen peroxide and then incubated with primary antibodies (10–40 min). The primary antibodies (Dako, Glostrup, Denmark) used in this study were CD34 monoclonal mouse anti-human, clone QBEnd-10 (dilution 1:50), catalog No. IR632 and α-smooth muscle actin (αSMA) monoclonal mouse anti-human, clone 1A4 (dilution 1:50), catalog No. IR611. The immunoreaction was developed in a solution of diaminobenzidine and the sections were then briefly counterstained with hematoxylin, dehydrated in ethanol series, cleared in xylene and mounted in Eukitt^®^ Sigma Aldrich, Saint Louis, MO, USA. Positive and negative controls were used. For the double immunostaining, we used anti-CD34 antibody (diaminobenzidine, DAB, as chromogen) to highlight CD34+ ECs and anti-αSMA (aminoethylcarbazole, AEC, substrate-chromogen) for anti-αSMA+ pericytes/smooth muscle cells.

Specimens for electron microscopy were initially fixed in glutaraldehyde solution, diluted to 2% with sodium cacodylate buffer, pH 7.4, for 6 h at 4 °C. Then they were washed in the same buffer, post-fixed for 2 h in 1% osmium tetroxide, dehydrated in a graded ethanol series, and embedded in epoxy resin. Serial semithin sections (1.5 μm) were mounted on acid-cleaned slides, stained with 1% Toluidine blue (Merck^®^, Darmstadt, Alemania) and observed under a Leica^®^, Wetzlar, Germany, DM-750 light microscope with an integrated High Definition Camera. Ultrathin sections were double-stained with uranyl acetate and lead citrate. The grids were examined at 60 kV with a JEOL^®^ 100B Akishima, Tokio, Japón, electron microscope.

## Figures and Tables

**Figure 1 ijms-21-08049-f001:**
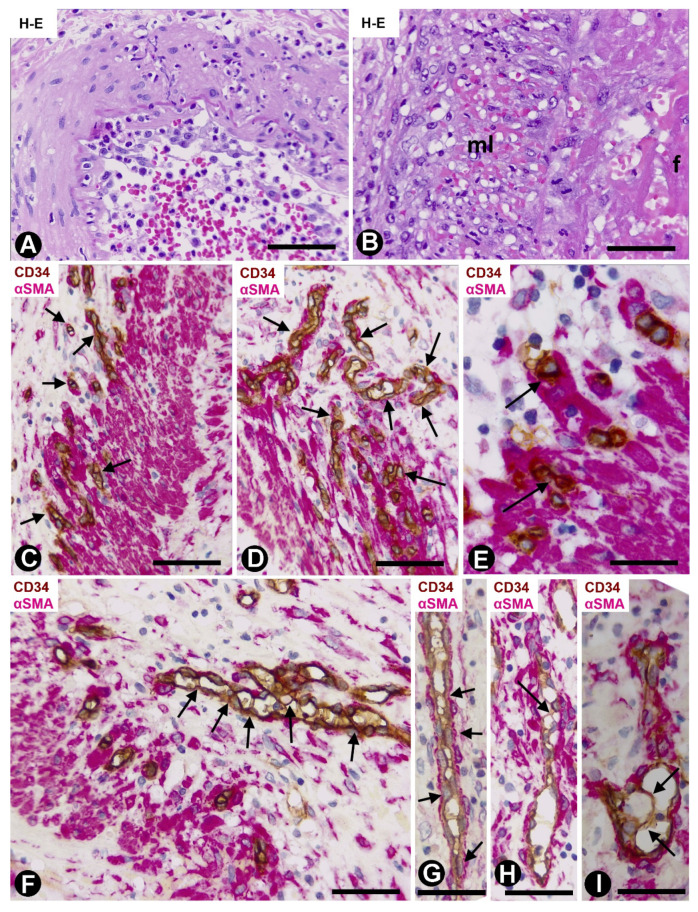
Arteries in gallbladders with acute cholecystitis and urgent cholecystectomy. Sections stained with Hematoxylin-Eosin (HE) (**A**,**B**) and double-immunostained with anti-CD34 (brown) and anti-αsmooth muscle actin αSMA (red) (**C**–**I**). (**A**) Arterial segment in which parietal and luminal infiltration of lymphocytes and neutrophils is observed. Note the loss of the intimal endothelium. (**B**) Presence of fibrin (f) in the arterial lumen and red blood cells in the media layer (ml). (**C**,**D**) Numerous microvessels (arrows) with endothelial cells (ECs) (brown) and pericytes (red) are seen in the adventitia. (**E**) Microvessels in the arterial media layer. Observe that the ECs (brown) seem to be surrounded by smooth muscle cells (arrows). (**F**–**I**) Presence of EC bridges and endothelial contacts in the adventitial microvessels (arrows). Bar: (**A**,**B**): 100 μm; (**C**,**D**,**F**–**I**): 80 μm; (**E**): 30 μm.

**Figure 2 ijms-21-08049-f002:**
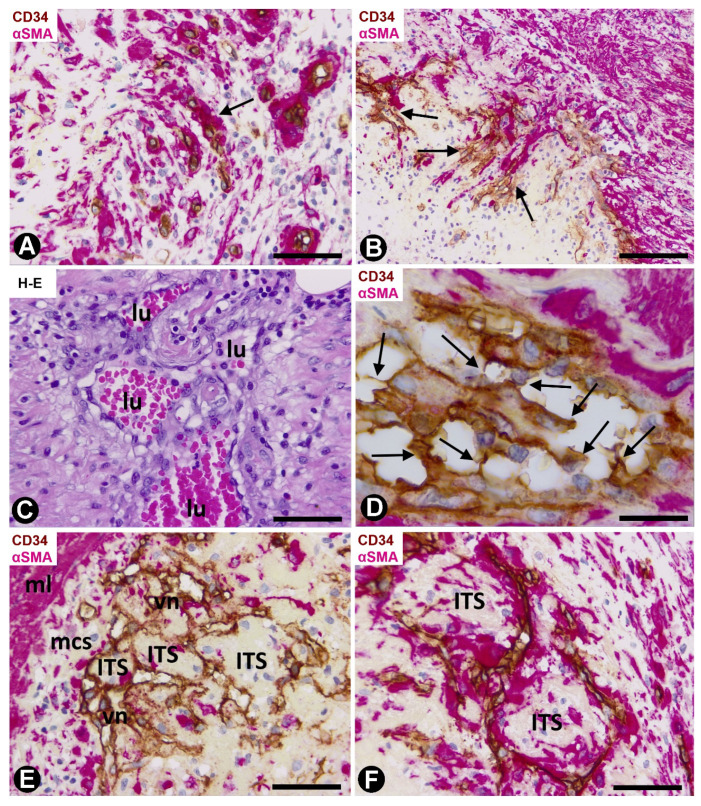
Arteries in gallbladders with acute cholecystitis and urgent cholecystectomy. Sections double-immunostained with anti-CD34 (brown) and anti-αSMA (red) (**A**,**B**,**D**–**F**) and stained with HE (**C**). (**A**) Numerous capillaries (arrows) in a cell-diminished artery media layer. (**B**) Endothelial cells (ECs) (brown) and vascular smooth muscle cells (VSMCs) (red) (arrows) in the arterial lumen. (**C**) Several lumens (lu) formed by ECs. (**D**) Presence of EC bridges forming arterial neolumens (arrows). (**E**) A complex vessel network (vn) is observed underlying the VSMCs (myointimal cells: mcs), which are under the muscular layer (ml) of the artery. Note the formation of loops, and ECs (brown) and αSMA+ cells in the neo-capillaries of the complex network. Interstitial tissue structures (ITSs) are also seen. (**F**) Detail of loops in the vessel networks. Note interstitial tissue structures (ITSs) surrounded by loops. Bar: (**B**): 100 μm; (**A**,**E**,**F**): 80 μm; (**C**): 70 μm; (**D**): 10 μm.

**Figure 3 ijms-21-08049-f003:**
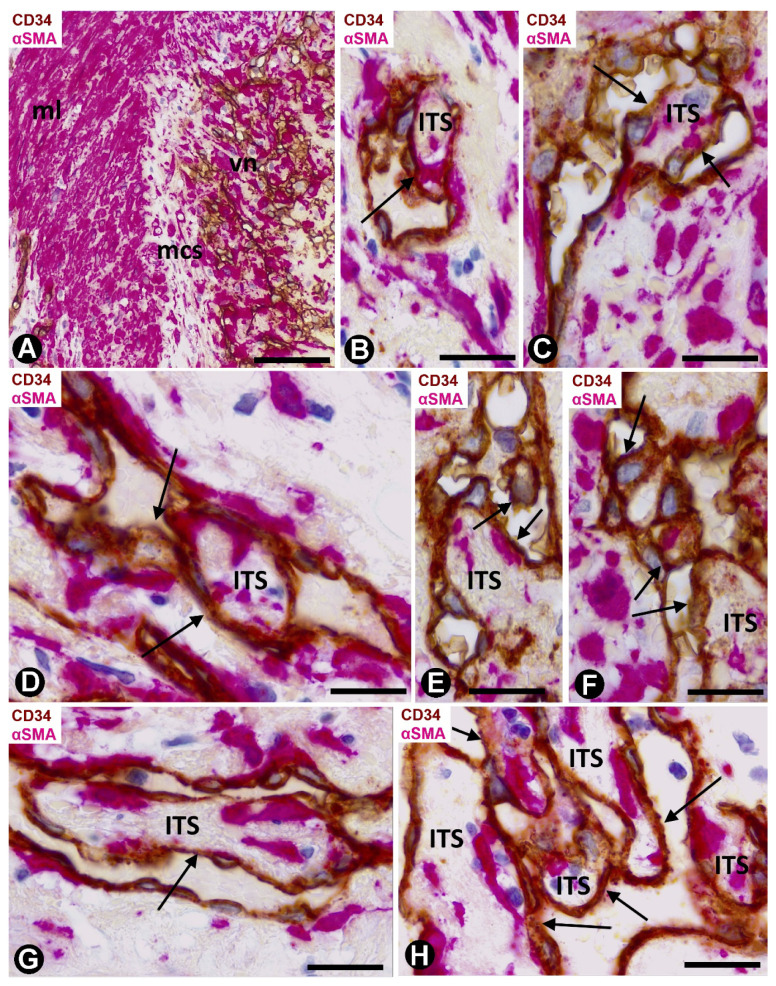
Arteries in gallbladders with acute cholecystitis and urgent cholecystectomy. Sections double-immunostained with anti-CD34 (brown) and anti αSMA (red). (**A**) Region with early intimal thickening formed by myointimal cells (mcs) and the underlying vessel network (vn) (ml: media layer). (**B**–**H**) Several images of vessel loops, ITSs and endothelial bridges, forming pillars (arrows). Note that ITSs containing cells expressing αSMA are surrounded by the inner layer of the loops and are partially or totally located in the vessel lumens. Occasionally, ITSs form complex structures (**H**). Bar: (**A**): 80 μm; (**B**–**H**): 10 μm.

**Figure 4 ijms-21-08049-f004:**
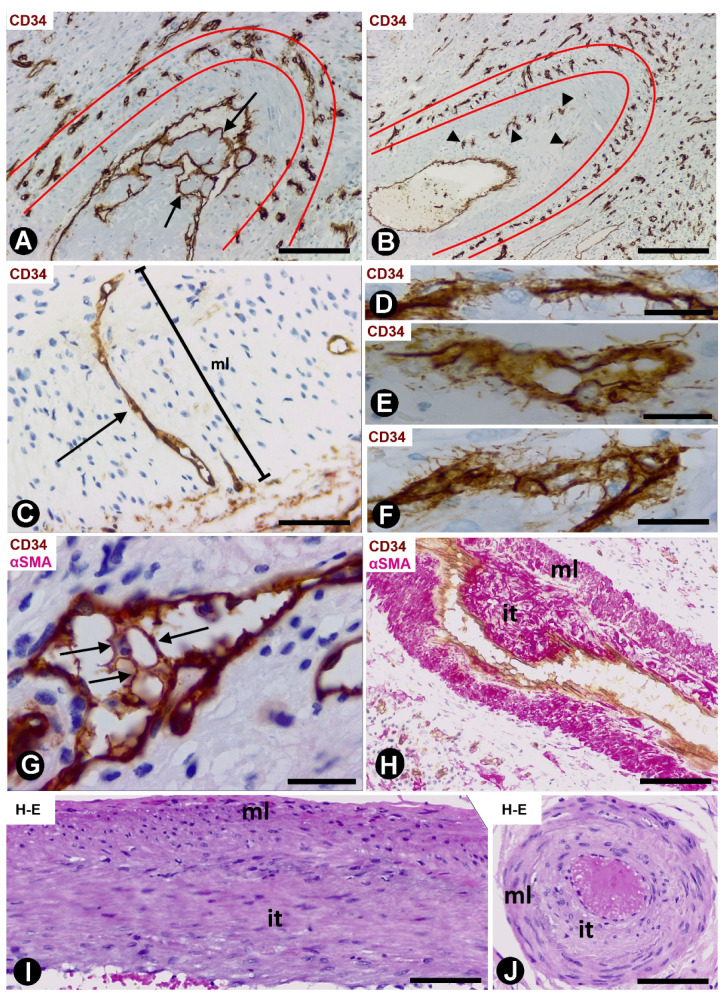
Arteries in gallbladders with acute cholecystitis and delay cholecystectomy. Sections immunostained with anti-CD34 (**A**–**F**), double-immunostained with anti-CD34 (brown) and αSMA (red) (**G**,**H**) and stained with HE (**I**,**J**). (**A**,**B**) arteries with numerous residual microvessels (delimited between red lines) in the adventitia. Some large pillars are observed in A (arrows) and residual microvessels (arrowheads) underlying the elastic interna in B. (**C**) A residual microvessel (arrow) is shown crossing the entire thickness of the arterial wall from the adventitia to the intimal thickening. (**D**–**F**) Numerous CD34+ filopodia-like are observed in ECs of some microvessels. (**G**) Intravascular bridges in residual vessels (arrows). (**H**–**J**) Presence of well-formed intimal thickening (it) (arterial media layer: ml). Bar: (**A**,**B**,**H**): 160 μm; (**C**): 80 μm; (**G**): 20 μm; (**D**–**F**): 10 μm; (**I**,**J**): 60 μm.

**Figure 5 ijms-21-08049-f005:**
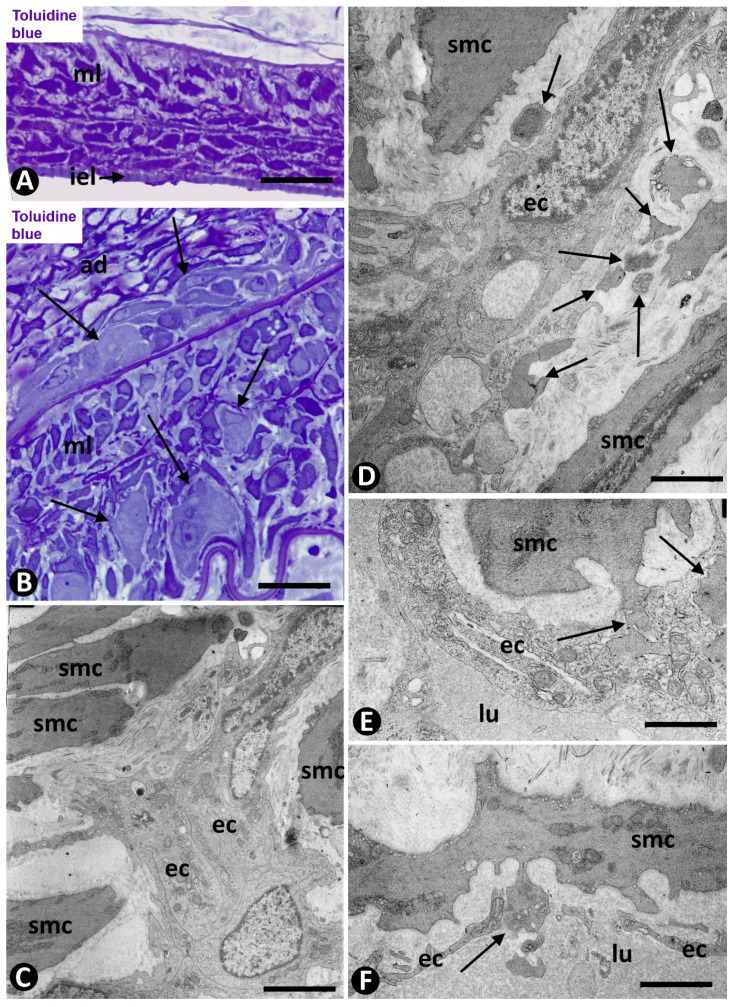
Angiogenesis in experimental occluded arterial segments. (**A**) Arterial segment with loss of endothelium (internal elastic lamina: iel; media layer: ml). (**B**) Capillaries (arrows) in the adventitia (ad) and media layer (ml) of an artery. (**C**–**F**) ECs (ec) of the capillaries are surrounded by VSMCs (smc) and globular structures (arrows). Note that the globular structures are widened terminal portions of cytoplasmic processes of VSMCs (**E**,**F**). These bulb-shaped terminal portions establish peg-and-socket-type heterocellular junctions with the ECs (**E**) and they even reach (arrow) the vessel lumen (**F**). Vessel lumen (lu). (**A**,**B**) Semithin sections stained with Toluidine blue. (**C**–**F**) Ultrathin sections, uranyl acetate and lead citrate. Bar: (**A**): 25 μm; (**B**): 10 μm; (**C**,**D**): 3 μm; (**E**,**F**): 2 μm.

**Figure 6 ijms-21-08049-f006:**
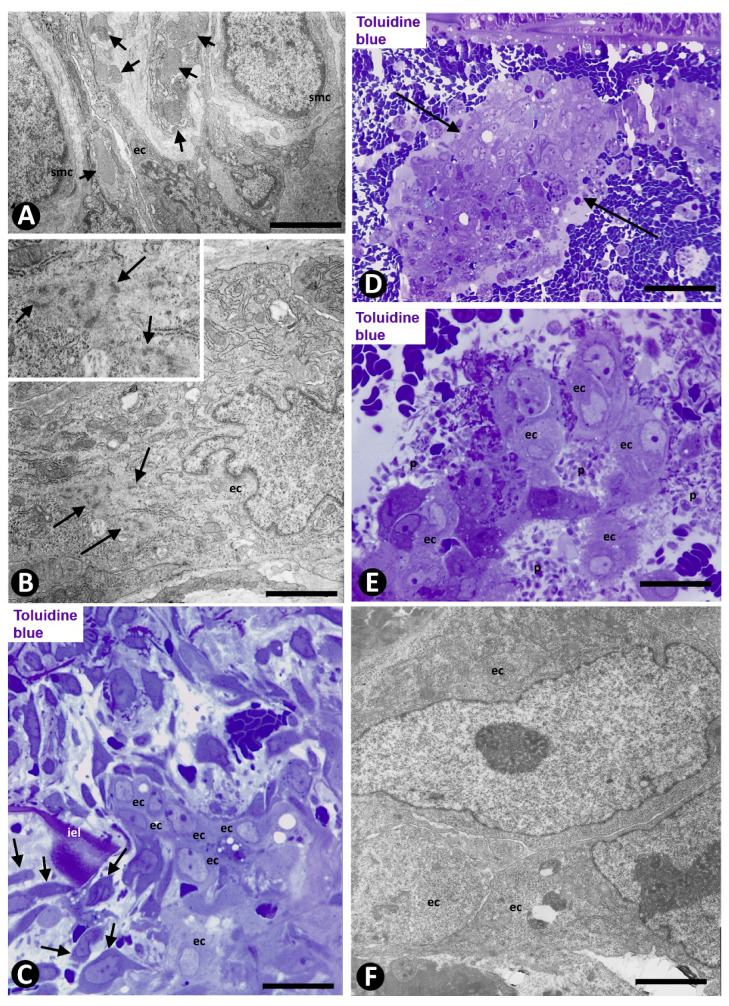
Angiogenesis in experimental occluded arterial segments. (**A**,**B**). Smooth muscle cells (SMCs) (smc) and ECs (ec) in capillaries with prominent nuclei and cytoplasms with abundant ribosomes. Numerous globular structures are seen in the interstitium and invaginated into Ecs (A, arrows). Stress fibers (arrows) are observed in ECs (B and insert). (**C**) One capillary is seen penetrating the arterial internal elastic lamina (iel). Note that VSMs (arrows) extend over the luminal surface of this lamina and that ECs (ec) form an aggregate. (**D**–**F**) Aggregates of ECs (arrows in D), between red blood cells and numerous platelets (p in E), in the arterial lumen. ECs (ec) are voluminous with large nuclei and prominent nucleoli (**D**–**F**). (**A**,**B**,**F**) Ultrathin sections. Uranyl acetate and lead citrate. (**C**–**E**) Semithin sections: Toluidine blue. Bar: (**C**,**E**): 10 μm; (**D**): 40 μm; (**A**): 4 μm; (**F**): 2 μm; (**B**): 1 μm.

**Figure 7 ijms-21-08049-f007:**
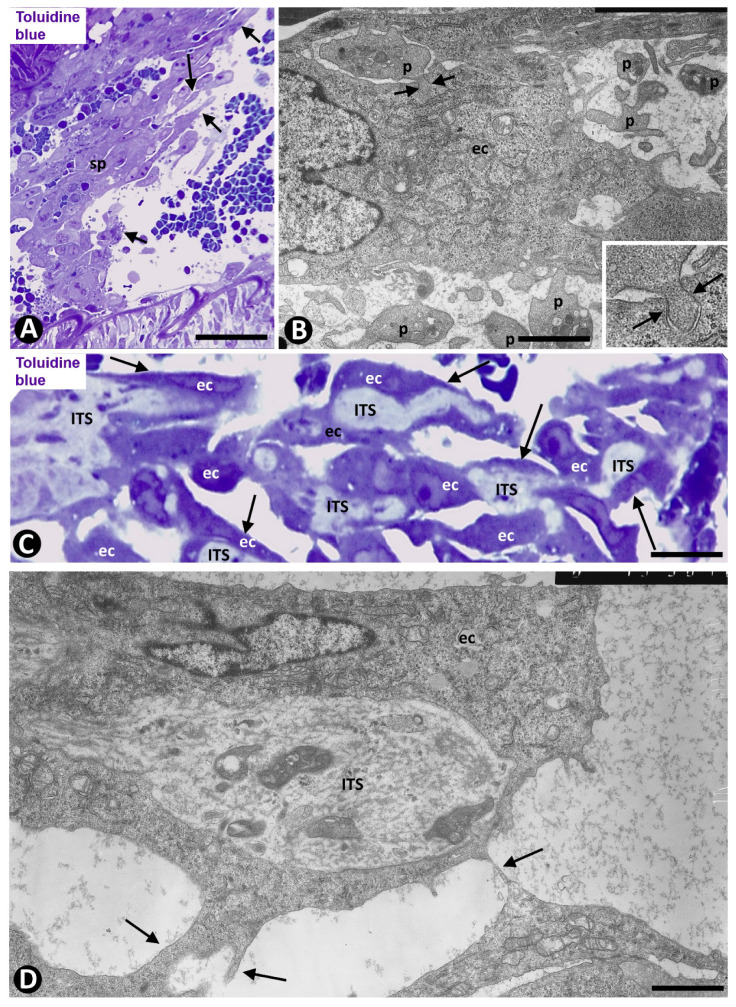
Angiogenesis in experimental occluded arterial segments. (**A**, **B** and insert of B) ECs extending from spheroids (sp) into the arterial lumen (arrows in A). In B and insert, note that platelets (p) associate with an EC (ec) and form peg-and-socket junctions (arrows) (Platelets form the peg and ECs form the socket). (**C**) Intraluminal pillars (arrows) covered by ECs (ec). Note the cores or ITSs formed in the spaces between primitive bilayers of ECs, whose abluminal surfaces face each other. (**D**) Ultrastructural image of a pillar formed by ECs (ec) (the cover) and an ITS (the core). Observe projections of ECs (nascent pillars-arrows) that join the pillar with other pillars (arrows). (**A**,**C**) Semithin sections. Toluidine blue. (**B**,**D**) Ultrathin sections. Uranyl acetate and lead citrate. Bar: (**A**): 40 μm; (**C**): 8 μm; (**B**,**D**): 3 μm.

**Figure 8 ijms-21-08049-f008:**
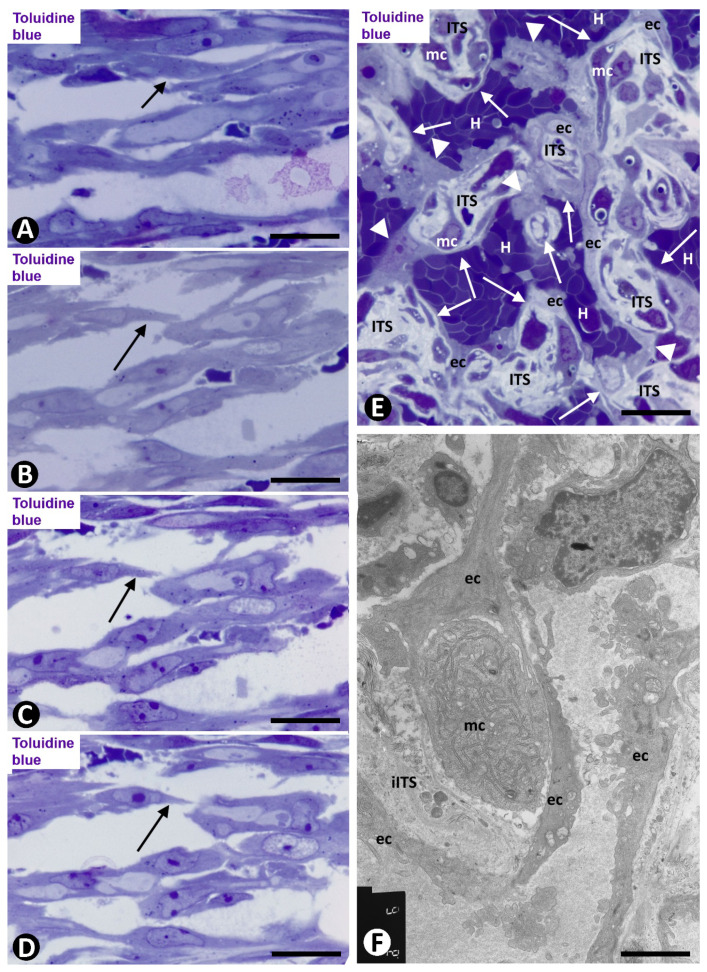
Angiogenesis in experimental occluded arterial segments. (**A**–**D**) Appearance/disappearance and continuities/discontinuities of the nascent and early pillars (e.g., arrows in a nascent pillar in successive sections) are observed in serial semithin sections. (**E**,**F**) Mature pillars (arrows) with myointimal cells (mc) in their ITSs. Note how the ECs (ec) that form the cover of pillars establish bridges with other pillars (arrowheads); (vessel lumen: lu; red blood cells: H). (**A**–**E**) Semithin sections. Toluidine blue. (**F**) Ultrathin section, uranyl acetate and lead citrate. Bar: (**E**): 10 μm; (**A**–**D**): 8 μm; (**F**): 3 μm.

**Figure 9 ijms-21-08049-f009:**
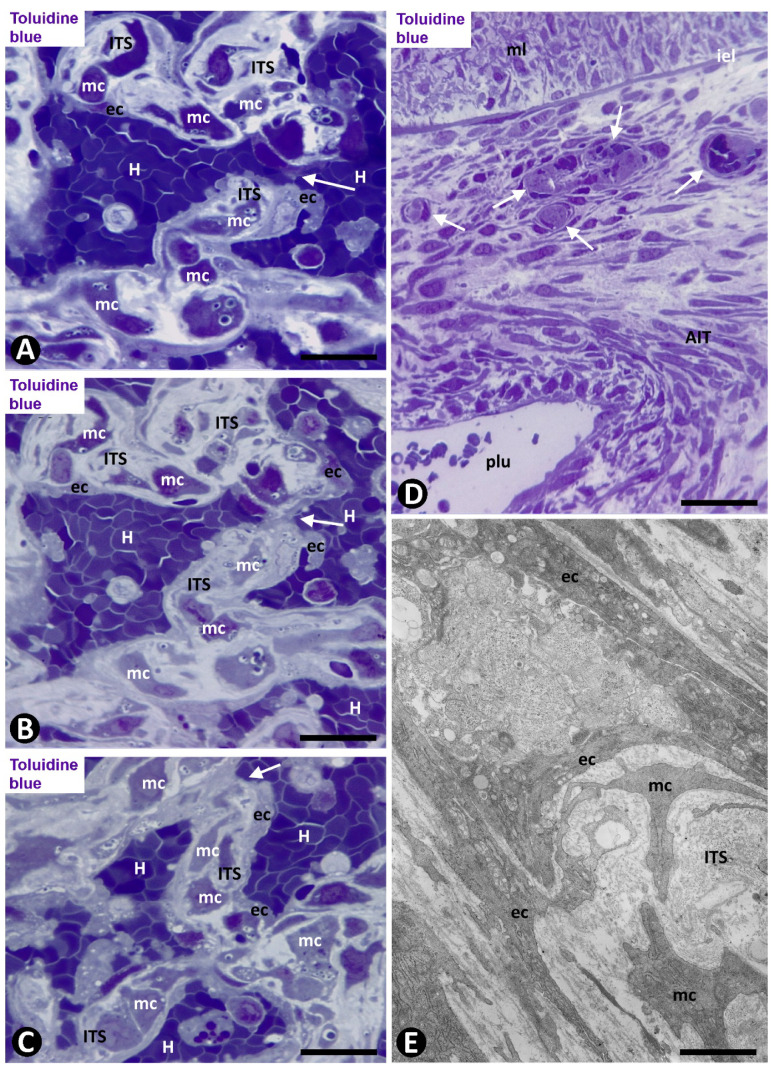
Angiogenesis, remodeling and vessel regression in experimental occluded arterial segments; mc: myointimal cells, ec: endothelial cells, H: red blood cells, ITS: interstitial tissue structure. (**A**–**C**) Appearance/disappearance and continuities/discontinuities of mature pillars in serial semithin sections (the arrows point to a pillar that contacts, or not, with another pillar, depending on the serial semithin section). (**D**) Artery with intimal thickening and a preferential lumen (plu). Note regressive vessels (arrows) within the intimal thickening. (**E**) Ultrastructural image of a regressive vessel and an ITS with persistent myointimal cells. (**A**–**E**) Semithin sections. Toluidine blue. (**F**) Ultrathin section. Uranyl acetate and lead citrate. Bar: (**D**): 30 μm; (**A**–**C**): 10 μm; (**E**): 2 μm.

**Figure 10 ijms-21-08049-f010:**
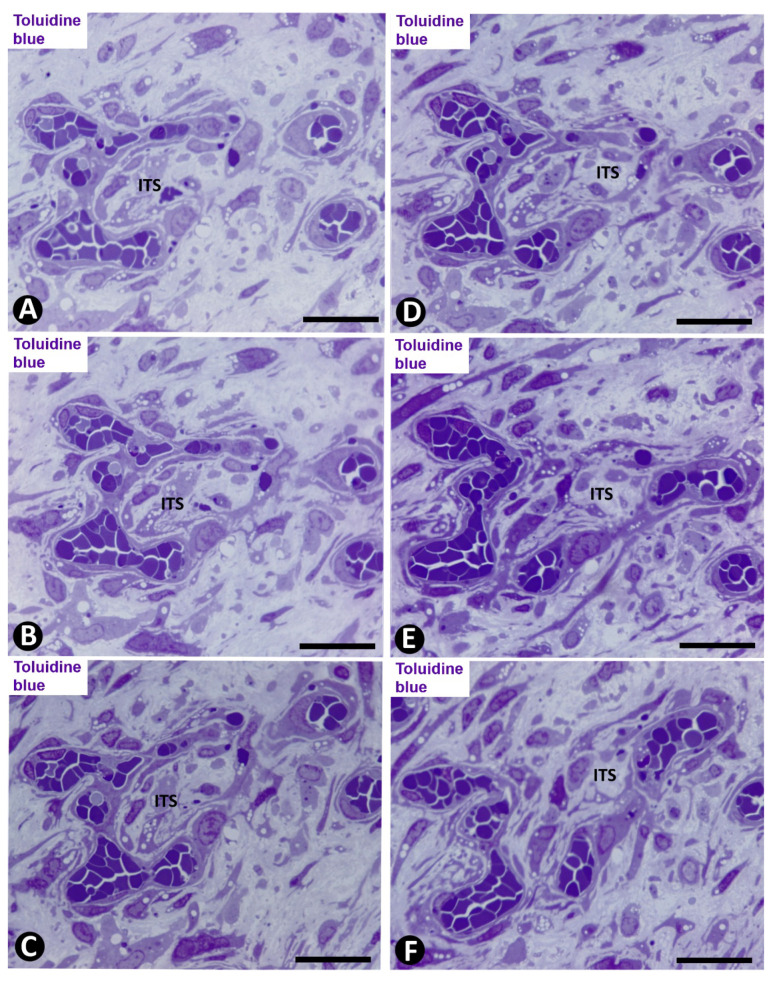
Intussusception with regression of a newly formed vessel loop in an occluded arterial segment. (**A**–**F**): Serial semithin sections, in which the appearance/disappearance of interendothelial contacts, contact perforations and fragmentation in regressive capillary-like structures are showed in the vessel loop. ITS: Interstitial tissue structure. Toluidine blue. Bar: (**A**–**F**): 30 μm.

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
