# Peer review of "Intussusceptive Angiogenesis and Peg–Socket Junctions between Endothelial Cells and Smooth Muscle Cells in Early Arterial Intimal Thickening"

_ijms, 2020, doi:10.3390/ijms21218049_

Round 1

Reviewer 1 Report

In this work Diaz-Flores and colleagues performed IHC and electron microscopy to characterize the interactions and relationships between endothelial cells (ECs ) and smooth muscle cells (SMC), as well as by including immune cells, in intussusceptive angiogenesis ongoing in human arteries of patients with vasculitis in gallbladders.

The authors used a very appropriate number of patients to perform their experiments. Data have been very well presented and discussed and the experiments have been properly performed. 

This reviewer has anyway some concerns that must be addressed from authors.

Since the authors' study and the conclusions indicated are based only on immunological data, therefore observational, authors tend to use terms such as "cell migration" or terms characterizing phenotypic changes (eg. neointimal or myointimal) too loosely. This turns out to be a little too speculative, considering that the histochemical data do not allow to draw conclusions on migratory states or phenotypic changes.

Since the study has a solid foundation in patient numbers and time points, this reviewer believes it is possible to include additional data that would make the relationship hypothesized here more solid. In particular:

1. ECs migration as well as interconnections should be proven by staining of junction proteins or activation markers which are characteristic of endothelial cell activation states. Authors could refer to the works of Carmeliet or other authors for markers to be used in IHC.

2. Linked to point 1., the same should be done for vSMC

3. Additional markers of micro and macrovasculature are available and would support the co-regulatory role hypothesized here, i.e. laminin, MEGA32, SCA-1 and Endomucin

4. Although the time points come (rightly) from different patients, the authors should include a correlation analysis.

Minor concerns:

5. The authors mention migrations and neo-vascularization of ECsm but how to exclude the mere damage deriving from tissue harvesting?

Author Response

Reviewer 1

1, 2, 3. The main limitation in our work is the difficulty of demonstrating endothelial cell and smooth muscle cell interconnections by staining junction proteins and markers to support a co-regulatory role in peg-and-socket junctions. These junctions were demonstrated by electron microscopy and are one of the contributions of our study. The specimens were conventionally processed for electron microscopy, which limits the detection of antigens in peg-and-socket junctions. This issue can be extended to the peg-and socket junctions between endothelial cells and platelets, a phenomenon not previously described and incidentally observed with relative frequency in our study. In the Discussion we have referenced this limitation as follows:

“The main limitations of this work are complementary studies in peg-and-socket junctions and accurate time points of the pathologic processes that affect human gallbladders. The first limitation is that peg-and-socket junctions were demonstrated ultrastructurally and that the conventional electron microscopy procedures used limit junction protein staining or that of other markers to identify their possible regulatory role……”

Moreover, because demonstrating intussusceptive angiogenesis and peg-and-socket junctions during arterial intimal thickening formation are the main contributions of our work, we have reviewed the work and withdrawn or simply suggested speculative conclusions, such as those on migratory states or phenotypic changes.

4 Another limitation is the time points of pathologic processes in human gallbladders deriving from different patients. To partially avoid this difficulty we selected cases with urgent and delayed cholecystectomy, which allowed us to observe intimal thickening in initial and more advanced stages. In addition, the findings in urgent and delayed cholecystectomy correlate with those obtained in experimental conditions, in which time points were well established. Exceptions were specific facts only observed by serial semithin section or electron microscopy. In the Discussion, we take also into consideration this limitation as follows:

“ ……...The other limitation was partially obviated by the selection of cases with urgent and delayed cholecystectomy, which allowed us to observe intimal thickening in the human gallbladder in initial and more advanced stages. In addition, there is a correlation of the findings in these stages with those obtained in experimental conditions in which the time points were well established”.

  1. We used controls from previous works, including unaffected human gallbladders and normal and sham-operated femoral arteries. We now include these controls. In Results we added:

“Arteries in unaffected human gallbladders were within normal limits”

“Normal and sham-operated femoral arteries were unmodified, except for the presence of a few neutrophils and mononuclear cells in the surrounding tissue of the sham-operated cases”.

Thank you for your recommendations, which have improved this paper.

Reviewer 2 Report

The present paper aims to assess the angiogenic events occurring in the initial stages of AIT development in medium- and small arteries, the possible involvement of intussusceptive angiogenesis during AIT development and the relationship between ingrowing ECs in the arterial media layer and VSMCs. Human arteries of gallbladders surgically removed for acute cholecystitis were explored, using conventional and immunohistochemical procedures, as well as occluded segments of rat femoral artery, using electron microscopy and serial semithin sections.

A few changes are needed, as follows:

Introduction: Please define intussusceptive and sprouting angiogenesis!

Please also include a definition of atherosclerosis: “the net effect of endothelial dysfunction, prothrombotic state and vascular inflammation resulting in plaque formation” (Tibaut M et al. Markers of Atherosclerosis: Part 1 - Serological Markers. Heart Lung Circ. 2019;28(5):667-677. doi: 10.1016/j.hlc.2018.06.1057)

Discussion: Please include study limitations and emphasize the importance of your findings for clinical practice!

Author Response

Reviewer 2

Sentences added to the Introduction:

1.  “Atherosclerosis, the pathologic process with major morbidity and mortality, has its morphological substrate in the atheromatous plaque, which leads to artery hardening and narrowing.  Plaque formation in atherosclerosis involves intertwined pathologic pathways and endothelial dysfunction, prothrombotic state and vascular inflammation (1,2)”.

2. “Sprouting angiogenesis is defined as a multistep complex process of neovascularization by the sprouting of capillaries from pre-existing vessels. This process includes EC migration, extracellular matrix changes, EC proliferation, pericyte mobilization, tubulogenesis (vascular lumen development), formation of a new basal membrane and sprouting connection. Inflammatory cells also participate in this process. Intussusceptive angiogenesis is the mechanism by which pre-existing vessels split, expand and remodel through transluminal pillar formation, contributing to microvascular growth, vessel arborization, branching remodelling and vessel segmentation (10)”.

Sentences added to the Discussion:

1. “The main limitations of this work are complementary studies in peg-and-socket junctions and accurate time points of the pathologic processes that affect human gallbladders. The first limitation is that peg-and-socket junctions were demonstrated ultrastructurally and that the conventional electron microscopy procedures used limit junction protein staining or that of other markers to identify their possible regulatory role. The other limitation was partially obviated by the selection of cases with urgent and delayed cholecystectomy, which allowed us to observe intimal thickening in the human gallbladder in initial and more advanced stages. In addition, there is a correlation of the findings in these stages with those obtained in experimental conditions in which the time points were well established”.

2. “The demonstrated participation of intussusceptive angiogenesis during AIT development may have clinical and therapeutic implications. Since sprouting and intussusception are complementary forms of angiogenesis, with synergistic interaction (10–16), a compensatory mechanism after inhibitory treatment of sprouting angiogenesis may occur due to intussusceptive angiogenesis (failure of treatment by an escape mechanism) (66). Likewise, the peg-and-socket junctions between ECs and smooth muscle cells and between ECs and platelets may be the morphologic substrate of mechanical and biochemical cellular interactions. Further studies are also required in these fields".

Thank you for your recommendations, which have improved this paper.